# Prognostic Impact of Immunoglobulin Kappa C (*IGKC*) in Early Breast Cancer

**DOI:** 10.3390/cancers13143626

**Published:** 2021-07-20

**Authors:** Marcus Schmidt, Karolina Edlund, Jan G. Hengstler, Anne-Sophie Heimes, Katrin Almstedt, Antje Lebrecht, Slavomir Krajnak, Marco J. Battista, Walburgis Brenner, Annette Hasenburg, Jörg Rahnenführer, Mathias Gehrmann, Pirkko-Liisa Kellokumpu-Lehtinen, Ralph M. Wirtz, Heikki Joensuu

**Affiliations:** 1Department of Obstetrics and Gynecology, University Medical Center Mainz, 55131 Mainz, Germany; anne-sophie.heimes@unimedizin-mainz.de (A.-S.H.); katrin.almstedt@unimedizin-mainz.de (K.A.); antje.lebrecht@unimedizin-mainz.de (A.L.); slavomir.krajnak@unimedizin-mainz.de (S.K.); marco.battista@unimedizin-mainz.de (M.J.B.); walburgis.brenner@unimedizin-mainz.de (W.B.); annette.hasenburg@unimedizin-mainz.de (A.H.); 2Leibniz Research Centre for Working Environment and Human Factors (IfADo) at Dortmund TU, 44139 Dortmund, Germany; Edlund@ifado.de (K.E.); Hengstler@ifado.de (J.G.H.); 3Department of Statistics, TU Dortmund University, 44221 Dortmund, Germany; rahnenfuehrer@statistik.tu-dortmund.de; 4Bayer AG, 42113 Wuppertal, Germany; mathias.gehrmann@bayer.com; 5Cancer Centre, Tampere University Hospital and University of Tampere, 33520 Tampere, Finland; Pirkko-Liisa.Kellokumpu-Lehtinen@tuni.fi; 6STRATIFYER, 50935 Köln, Germany; ralph.wirtz@stratifyer.de; 7Department of Oncology, Helsinki University Hospital and University of Helsinki, 00290 Helsinki, Finland; Heikki.Joensuu@hus.fi

**Keywords:** triple-negative breast cancer, prognosis, immune system, immunoglobulin kappa C

## Abstract

**Simple Summary:**

We examined the relevance of immunoglobulin kappa C (*IGKC*), an important part of the humoral immune system, in early breast cancer. To our knowledge, our results confirm for the first time previous retrospective findings of a cancer recurrence protective role of *IGKC* in a large cohort of early breast cancer patients who were treated in the prospective, randomized FinHer clinical trial. We show that an increased amount of *IGKC* in the tumor is linked to longer distant metastasis-free survival, especially in patients whose breast cancer does not express hormone receptors or human epidermal growth factor receptor-2. This type of breast cancer often has poor prognosis. Since an improved outcome is associated with the presence of tumor-infiltrating *IGKC* expressing immune cells, this may be a further argument for the use of immunotherapies in these patients.

**Abstract:**

We studied the prognostic impact of tumor immunoglobulin kappa C (*IGKC*) mRNA expression as a marker of the humoral immune system in the FinHer trial patient population, where 1010 patients with early breast cancer were randomly allocated to either docetaxel-containing or vinorelbine-containing adjuvant chemotherapy. HER2-positive patients were additionally allocated to either trastuzumab or no trastuzumab. Hormone receptor-positive patients received tamoxifen. *IGKC* was evaluated in 909 tumors using quantitative real-time polymerase chain reaction, and the influence on distant disease-free survival (DDFS) was examined using univariable and multivariable Cox regression and Kaplan–Meier estimates. Interactions were analyzed using Cox regression. *IGKC* expression, included as continuous variable, was independently associated with DDFS in a multivariable analysis also including age, molecular subtype, grade, and pT and pN stage (HR 0.930, 95% CI 0.870–0.995, *p* = 0.034). An independent association with DDFS was also found in a subset analysis of triple-negative breast cancers (TNBC) (HR 0.843, 95% CI 0.724–0.983, *p* = 0.029), but not in luminal (HR 0.957, 95% CI 0.867–1.056, *p* = 0.383) or HER2-positive (HR 0.933, 95% CI 0.826–1.055, *p* = 0.271) cancers. No significant interaction between *IGKC* and chemotherapy or trastuzumab administration was detected (P_interaction_ = 0.855 and 0.684, respectively). These results show that humoral immunity beneficially influences the DDFS of patients with early TNBC.

## 1. Introduction

During the last decade, numerous, largely retrospective, analyses showed that tumor-infiltrating lymphocytes or transcripts of immune cells play an important prognostic and predictive role in breast cancer. We have previously reported a strong beneficial prognostic impact of T cell as well as B cell metagenes for breast cancer prognosis [1]. The strong protective impact of a B cell/plasma cell signature were later confirmed by others [2,3]. Tumor-infiltrating plasmablasts and plasma cells were identified as the source of immunoglobulin kappa C (*IGKC*) expression using confocal microscopy [4]. In this study, co-staining with anti-human IgG showed that *IGKC* was expressed in IgG-positive cells, a well-known feature of B cell maturation and plasma cell differentiation after antigen encounter. *IGKC* was associated with favorable prognosis in patients who had not been treated with systemic therapy, and also with response to anthracycline-containing neoadjuvant chemotherapy in early breast cancer. Indeed, Gentles et al. confirmed that plasma cell signatures and also plasma cells expressing *IGKC* were associated with improved survival in a comprehensive analysis of the prognostic landscape of genes and infiltrating immune cells across human cancers [3]. Taken together, these and other results suggest that humoral immunity might be as important as cellular immunity in eliminating cancer [5]. For example, B cell-attracting C-X-C motif chemokine ligand 13 (CXCL13)-positive, CD4-positive T follicular helper T (Tfh) cells were independently associated with distant disease-free survival (DDFS) in patients with triple-negative breast cancer (TNBC) in the FinHer trial patient population [6]. Recently, Garaud and co-workers examined tumor-infiltrating B cells (TIL-B) in TNBC from the BIG 02-98 clinical trial and showed a correlation between Tfh TILs and antibody secretion [7]. 

Even though Tfh cells play an important role in humoral immune responses, antibody-secreting immune cells are the most obvious and definite proof of humoral immunity. A strong association between *IGKC*, tumor-associated plasma cell infiltration and improved prognosis was found by Yeong and co-workers in a retrospective cohort of 269 TNBC samples [8]. However, all of these results were obtained analyzing retrospectively collected breast cancer samples, which carries the risk of a selection bias. Thus, in order to avoid the potential biases that may arise in such retrospective series, we studied the prognostic impact of *IGKC* in the patient population of the prospective FinHer trial that evaluated the inclusion of vinorelbine and trastuzumab to the standard adjuvant treatments in early breast cancer [9]. To our knowledge, the results presented here are the first to confirm an independent prognostic impact of *IGKC* in patients with early TNBC in an exploratory analysis of a large randomized trial.

## 2. Materials and Methods

In the FinHer trial (International Standard Randomized Controlled Trial number, ISRCTN76560285) 1010 node-positive or high-risk node-negative breast cancer patients were randomized. Patients received either three cycles of adjuvant docetaxel or vinorelbine, followed by three cycles of fluorouracil, epirubicin, and cyclophosphamide [9]. Patients with human epidermal growth factor receptor 2 (HER2)-positive cancer were additionally randomized between nine trastuzumab infusions administered at one-week intervals concomitantly with chemotherapy (either with vinorelbine or docetaxel) and no trastuzumab. Steroid hormone receptor-positive patients received tamoxifen. The patients were recruited from 17 study centers from October 2000 to September 2003. One patient was excluded from the analysis due to presence of overt distant metastases already at the time of study entry (Figure 1).

Ethical approval was obtained from an ethics committee at the Helsinki University Central Hospital. Study participants provided signed informed consent to allow further research analyses on their tumor tissue. This exploratory biomarker study is reported according to the REMARK (Reporting Recommendations for Tumor Marker Prognostic Studies) criteria [10]. The characteristics of the patients and the tumors are shown in Table 1.

Determination of steroid hormone receptor status and HER2 expression by immunohistochemistry (IHC) was performed according to the guidelines of each institution [9]. When HER2 expression was scored 2+ or 3+, the number of copies of HER2 was centrally confirmed by means of chromogenic in situ hybridization (CISH) [11]. The cancers were considered hormone receptor positive when ≥10% of cancer cells expressed estrogen receptor (ER) and/or progesterone receptor PR). Patients with ER-positive or PR-positive tumor were scheduled to receive 5 years of tamoxifen. Breast cancer subtypes were classified using IHC as previously described as either luminal (ER+ and/or PR+, HER2−), HER2-positive (HER2+, irrespective of the steroid hormone receptor status), or TNBC (ER−, PR−, HER2−) [12]. 

Total RNA was extracted from 5-μm thick tumor formalin-fixed paraffin-embedded (FFPE) tissue sections with ≥30% of the section surface area consisting of cancer as previously described [6]. In 950 (94.1%) out of the 1010 cases, tumor tissue was available for RNA extraction. *IGKC* could be successfully analyzed from 910 (90.0%) tumors. After xylene-free deparaffinization driven by heat, the RNA extraction was done using a commercially available kit (XTRAKT-R02; STRATIFYER, Cologne, Germany). The DNase digestion was done using Ambion RNase-free DNase I. The same quality measures were applied as were previously used for the successful clinical development of RNXtract^®^ and MammaTyper^®^ IVD [13,14]. Each extraction was checked for mRNA quantity and amplificability by using clinically developed reference genes demonstrating sufficient fragment length of total RNA extract for the amplification. Samples with low mRNA content (raw CT value of CALM2 > 30) were excluded from analysis. DNA content after DNase digestion was checked by PCR using DNA specific, intron-spanning PAEP gene as quality control gene [15]. Assay performance and specificity of the *IGKC* assay was validated by using commercially available universal human reference RNA (Stratagene, La Jolla, CA, USA) and human genomic DNA (Roche Diagnostics, Mannheim, Germany). The RNA quality was assessed by the ability to amplify a 65 bp fragment of *RPL37A* and the Cq value was used as a surrogate marker for the mRNA yield, as described in Bohmann et al. [16]. One-step qRT-PCR to measure tumor *IGKC* content was done using a custom-designed gene-specific TaqMan-based assay. *IGKC* and the reference gene *CALM2* mRNA content were assessed in triplicates using the SuperScript III Platinum One-Step Quantitative RT-PCR System with ROX (Invitrogen, Karlsruhe, Germany) in a Versant kPCR system (Siemens, Erlangen, Germany). The thermal profile included 30 min at 50 °C, 20.5 min at 8 °C, and 2 min at 95 °C followed by 40 cycles of 15 s at 95 °C, and 30 s at 60 °C. Forty amplification cycles were applied and the cycle quantification threshold (Ct) values of *IGKC* and one reference gene for each sample were estimated as the median of the triplicate measurements. The primer and probe sequences used for *IGKC* mRNA quantification were for the probe AGCAGCCTGCAGCCTGAAGATTTTGC, the forward primer was GATCTGGGACAGAATTCACTCTCA and the reverse primer GCCGAACGTCCAAGGGTAA. For *CALM2*, the probe TCGCGTCTCGGAAACCGGTAGC, forward primer GAGCGAGCTGAGTGGTTGTG and reverse primer TGTGGTTCCTGCATGAAGACA were used. For *RPL37A*, the probe TGGCTGGCGGTGCCTGGA, forward primer TGTGGTTCCTGCATGAAGACA and reverse primer GTGACAGCGGAAGTGGTATTGTAC were used. The relative mRNA expression levels of *IGKC* were calculated as delta cycle threshold (ΔCt) values ΔCt = 40 − [Ct GOI (mean of gene of interest) − Ct REF (mean of *CALM2*)]. The final expression values were generated by subtracting ΔCT from the total number of cycles (40 – ΔCT) to ensure that the normalized gene expression obtained by the test was proportional to the corresponding mRNA expression.

The primary objective of this explorative study was to evaluate the association of tumor *IGKC* content with distant disease-free survival (DDFS), which was the survival endpoint in the final analysis of the FinHer trial [9]. The secondary objectives were to study the influence of *IGKC* expression in defined breast cancer molecular subtypes (i.e., luminal, HER2-positive, triple-negative), and the associations with the type of adjuvant therapy administered. DDFS was defined as the time interval between the date of randomization and the date of first cancer recurrence outside of the ipsilateral locoregional region or the date of death, whenever death occurred before distant recurrence. Patients alive without documented evidence of distant metastases were censored at the time of the latest contact. DDFS rates were determined using the Kaplan-Meier analyses. The log-rank test was used to compare survival between groups. We stratified the samples using the median as well as the top quartile *IGKC* mRNA expression as the cut-off value for each molecular subgroup; 66 triple-negative cancers had high and 66 low *IGKC* expression (median, 35.76), 289 luminal cancers had high and 285 low *IGKC* expression (median, 34.29), and 102 HER2-positive cancers had high and 101 low *IGKC* expression (median, 35.07). Frequency tables were analyzed using the Fisher’s exact test.

Univariable and multivariable Cox proportional hazards models were fitted to investigate the association of tumor *IGKC* expression as a continuous variable, as well as dichotomized using either the median or the top quartile as cutoff, with DDFS for each of the three molecular subtypes separately as well as combined. Other co-variables in the multivariate models were age at the time of study entry (≤50 vs. >50 years), breast tumor size (pT1 vs. pT2-4), axillary nodal status (pN0 vs. pN1-3), histological grade of differentiation (grade I vs. II-III for the entire cohort and the luminal subgroup, and I + II vs. III for the TNBC- and HER2-positive subgroups, respectively, due to the small number of grade I tumors in these subgroups), HER2 status (positive vs. negative), ER status (positive vs. negative), PR status (positive vs. negative), and molecular subtype (luminal vs. HER2-positive vs. triple-negative). A Cox proportional hazards model containing the treatment group (docetaxel vs. vinorelbine, or trastuzumab vs. no trastuzumab when the tumor was HER2-positive), *IGKC* expression (high [≥median] vs. low [<median]), and the treatment-by-biomarker interaction term was used to study the potential interactions between tumor *IGKC* content and the treatment assigned. 

All *p* values were two-sided, and *p* < 0.05 was considered statistically significant. All analyses were explorative and not adjusted for multiple testing. Therefore, *p* values should be interpreted with caution and in connection with the effect estimates. Statistical analyses were performed using the Statistical Package for Social Science (SPSS) (SPSS Inc., version 27, Chicago, IL, USA) and the statistical programming language R version 4.0.3.

## 3. Results

### 3.1. IGKC mRNA Expression Depends on the Molecular Subtype

Of the 909 cancers, 574 (63.1%) were luminal, 203 (22.3%) HER2-positive, and 132 (14.5%) triple-negative. *IGKC* expression was significantly associated with molecular subtype (TNBC vs. luminal, *p* < 0.001; TNBC vs. HER2-positive, *p* = 0.327; HER2-positive vs. luminal, *p* < 0.001). Triple-negative cancers showed the highest expression of *IGKC* (median 35.76; interquartile range [IQR] 33.89–37.52) followed by HER2-positive tumors (median 35.07; IQR 33.73–36.82) and luminal cancers (median 34.29; IQR 33.05–35.58) (Figure 2).

### 3.2. IGKC Expression Is Associated with Adverse Histopathological Characteristics

First, we examined an association between cancer *IGKC* expression (<median vs. ≥median) and clinicopathological characteristics using Fisher’s exact test. High levels of *IGKC* were significantly associated with poor histological grade of differentiation (*p* < 0.001), negative ER (*p* < 0.001) and PR (*p* < 0.001) status, positive HER2 status (*p* = 0.013), high (>20%) Ki-67 expression (*p* < 0.001), and HER2-positive subtype as well as TNBC molecular subtype (*p* < 0.001). Tumor *IGKC* mRNA content was not associated with age at diagnosis (*p* = 0.688), tumor size (*p* = 0.356), axillary nodal status (*p* = 0.113), surgery of the breast (*p* = 0.252) or lymph node dissection (*p* = 0.857) (Table 1). Radiotherapy was given according to each institution’s guidelines; it was mandated after breast-conserving surgery and was given to most patients with node-positive disease.

### 3.3. Association between IGKC Expression and Distant Disease-Free Survival

In a univariable Cox analysis, tumor *IGKC* expression did not show a significant association with DDFS (HR 0.982, 95% CI 0.920–1.048, *p* = 0.589); whereas, the standard prognostic factors were as expected significantly associated with DDFS (Appendix A). In a multivariable analysis of the whole series, *IGKC* was significantly associated with DDFS (HR 0.930, 95% CI 0.870–0.995, *p* = 0.034) when included as continuous variable (Table 2). Furthermore, in the multivariable analysis, cancer ER expression, PR expression, HER2 expression, Ki-67 expression, pT stage, pN stage, histological grade of differentiation, and the molecular subtypes were also independently associated with DDFS. When we instead stratified the patients using the median *IGKC* mRNA expression as the cut-off value, *IGKC* did not reach independent significance (HR 0.815, 95% CI 0.580–1.145, *p* = 0.238) (Appendix A). However, using the upper quartile, *IGKC* retained its independent significance for DDFS (HR 0.619, 95% CI 0.410–0.934, *p* = 0.022) (Appendix A).

### 3.4. Significance of IGKC in Triple-Negative Breast Cancer

We next analyzed the association between *IGKC* expression and DDFS in each molecular subtype separately. In TNBC, *IGKC* was not significantly associated with DDFS in the univariable Cox analysis when included as a continuous variable (HR 0.907, 95% CI 0.806–1.022, *p* = 0.109) (Table 3). However, when using the median as a cut-off, high *IGKC* expression was associated with favorable DDFS (HR 0.418, 95% CI 0.198–0.882, *p* = 0.022) (Appendix A) (Figure 3a). This association was even stronger when the top quartile was used to stratify the patients into groups with high and low *IGKC* expression (HR 0.172, 95% CI 0.041–0.719, *p* = 0.016) (Appendix A) (Figure 3b).

In the multivariable analysis of TNBC, adjusted for the key prognostic clinical parameters age, pT stage, pN stage, and grade, *IGKC* expression was independently associated with DDFS as a continuous variable (HR 0.843, 95% CI 0.724–0.983, *p* = 0.029) (Table 3) and also when using the median (HR 0.322, 95% CI 0.146–0.712, *p* = 0.005) or top quartile (HR 0.197, 95% CI 0.045–0.852, *p* = 0.030) as cutoff to define high and low *IGKC* expression (Appendix A). *IGKC* expression was not significantly associated with DDFS in the uni- or multivariate analysis in the luminal subtype (Appendix A) (Figure 3c,d) nor in the HER2-positive subtype (Appendix A) (Figure 3e,f).

### 3.5. Cancer IGKC mRNA Content Shows No Association with Systemic Treatment Effects

An examination of the potential interactions between *IGKC* expression and the systemic treatments given with DDFS revealed no significant interaction with the type of chemotherapy or whether or not trastuzumab was administered (P_interaction_ = 0.855 and 0.684, respectively).

## 4. Discussion

In this retrospective analysis of a prospective trial, breast cancer *IGKC* mRNA content was associated with DDFS in the total cohort of patients treated in the FinHer trial in multivariable but not in univariable analysis. When we analyzed the molecular subtypes separately, cancer *IGKC* content was independently associated with longer DDFS in TNBC, while it was not significant in luminal or HER2-positive breast cancer. 

The lack of an association between cancer *IGKC* expression and survival differs from our previous study in cohorts of node-negative breast cancer patients who did not receive systemic therapies [4]. A possible explanation for this is that the prognostic effect of cancer *IGKC* content may be obscured by the effects of the systemic cancer treatments administered, since all hormone receptor-positive patients in the FinHer trial received chemotherapy followed by tamoxifen. Besides chemotherapy, tamoxifen may also have immune-modulatory effects, potentially interfering with the prognostic effects of tumor-infiltrating immune cells, as anticipated when we developed a prognostic gene expression signature for ER-positive, HER2-negative patients treated with adjuvant endocrine therapy (EndoPredict™) [17]. In fact, not a single gene from our previously defined B cell metagene was selected for the EndoPredict™ test [17]. In breast cancer patients who did not receive adjuvant systemic therapy, a strong prognostic effect of a B cell metagene including *IGKC* was seen in patients with rapidly proliferating, node-negative, ER-positive, HER2-negative breast cancer [1]. In contrast, this prognostic effect was not observed in the present study investigating *IGKC* in ER-positive, HER2-negative patients treated with adjuvant tamoxifen.

Moreover, we recently reported a significant interaction between the prognostic impact of *IGKC* and tamoxifen treatment in a retrospective and non-randomized analysis of patients treated more than two decades ago with adjuvant chemotherapy with or without tamoxifen [18]. The positive prognostic impact of high cancer *IGKC* expression was most pronounced in patients who did not receive tamoxifen as endocrine treatment compared with those breast cancer patients treated with tamoxifen. The interaction test confirmed a significant interaction between tamoxifen treatment and the prognostic impact of *IGKC* expression (P_interaction_ = 0.04). Indeed, there is a growing body of evidence demonstrating immunomodulatory effects of tamoxifen [19]. These authors proposed that tamoxifen leads to a shift away from Th1 to Th2 immunity. In addition, Li and co-workers profiled differentially expressed intratumoral cytokines as a signature to evaluate the immune-polarizing side effects of tamoxifen. They could show that patients with low immune-polarizing side effects of tamoxifen (low Th2 polarization) had a lower risk of distant metastasis in a cohort of 608 breast cancer patients. In addition, in vitro data revealed that tamoxifen impaired differentiation of dendritic cells and reduced their immunostimulatory capacity [20]. These authors even speculated that tamoxifen may depress immunity and potentially interfere with immunotherapeutic strategies to improve antitumor immunity in breast cancer patients. Furthermore, a recent exploratory analysis of a prospective trial showed that CD8+ TILs have a considerably stronger prognostic impact in ER-positive patients not treated with tamoxifen compared to patients treated with tamoxifen [21]. The interaction test between CD8 status and tamoxifen treatment for relapse-free interval showed a trend (P_interaction_ = 0.082). 

Taken together, the proposed interaction between tamoxifen and the immune response may explain why *IGKC* loses its prognostic relevance in patients treated with tamoxifen. On the other hand, one has to consider the relationship of immune-related markers such as *IGKC* with trastuzumab. Currently, it is well accepted that the host immune system contributes significantly to trastuzumab efficacy [22]. Supporting this, the baseline percentage of TILs was not only associated with pathologic complete response (pCR) but also provided independent prognostic information in patients treated with trastuzumab/pertuzumab-based neoadjuvant chemotherapy [23]. Conversely, Loi and co-workers failed to find a significant association between TILs and distant disease-free survival of HER2-positive patients in the FinHer study [24]. This is in line with our findings for *IGKC* in HER2-positive FinHer patients. Instead, they detected a statistically significant interaction between higher TILs and increased trastuzumab benefit in HER2-positive disease (DDFS P_interaction_ = 0.025). However, this positive association between TILs and benefit from trastuzumab is disputable. Perez et al. reported conflicting results with respect to an association between TILs and trastuzumab benefit in HER2-positive patients randomized to trastuzumab or no trastuzumab within the N9831 trial [25]. A meta-analysis in early HER2-positive breast cancer showed that high baseline TILs were associated with increased pCR probability [26]. However, this meta-analysis failed to show an interaction between TILs and response to trastuzumab. Consistent with these results, we could not detect a significant interaction of *IGKC* and trastuzumab benefit.

Using the B cell/plasma cell associated transcript *IGKC*, we could confirm the beneficial prognostic effects of tumor-infiltrating immune cells in TNBC. Our results in this particular molecular subtype support previous findings [24,27,28,29,30] of an independent prognostic association of TILs with improved survival in TNBC. Considering that the total mutational burden is highest in TNBC, this significant association of tumor-infiltrating immune cells and TNBC is not surprising [31]. In addition, these authors described that the mutational burden was highly correlated with the neoepitope load (R2 = 0.86). A higher neoepitope load renders immunotherapy more efficacious. Indeed, clinical results using immune checkpoint inhibition with monoclonal antibodies against programmed cell death protein 1 (PD-1) or its ligand programmed cell death 1 ligand 1 (PD-L1) in patients with early as well as advanced TNBC showed encouraging results [32,33,34,35]. Conversely, and consistent with our findings of a lesser role for the immune system in luminal breast cancer, a recent randomized trial in advanced hormone receptor-positive patients showed that the addition of the PD-1 antibody pembrolizumab to eribulin did not improve survival compared with eribulin alone [36].

A potential limitation of our study is that it is exploratory and that the different molecular subtypes received different therapies in addition to the randomized adjuvant chemotherapy (e.g., tamoxifen in hormone receptor-positive and trastuzumab in HER2-positive subtypes, respectively) potentially obscuring the prognostic effect of *IGKC* in these molecular subtypes. A strength of the study, however, is that we report the prognostic significance of *IGKC* in a large, randomized trial with approximately 90% of the formalin-fixed, paraffin-embedded tumor tissue available for the *IGKC* mRNA analysis.

## 5. Conclusions

We confirm the independent prognostic significance of cancer *IGKC* content in a prospective-retrospective study. A multivariate analysis supported the favorable DDFS associated with high cancer *IGKC* content in the subset of TNBC patients treated in the FinHer trial. This highlights the importance of the humoral immune system in early TNBC.

## Figures and Tables

**Figure 1 cancers-13-03626-f001:**
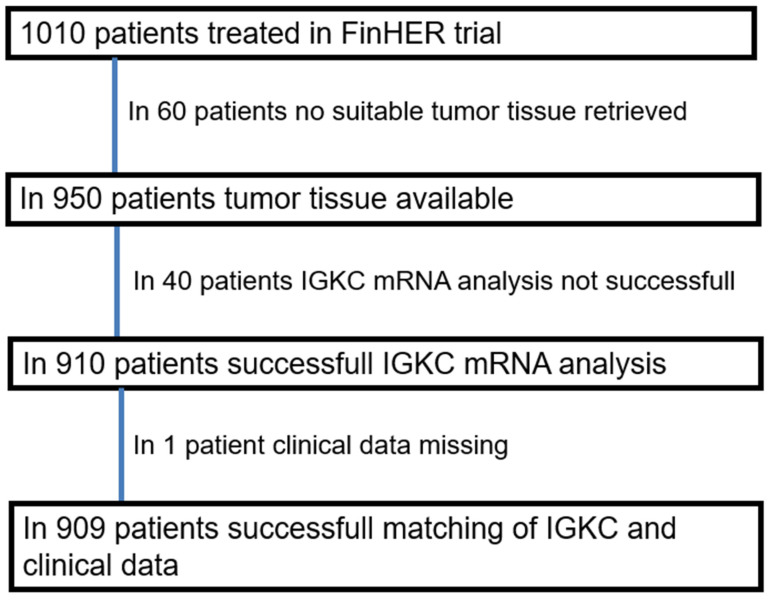
CONSORT diagram showing patient selection for the study.

**Figure 2 cancers-13-03626-f002:**
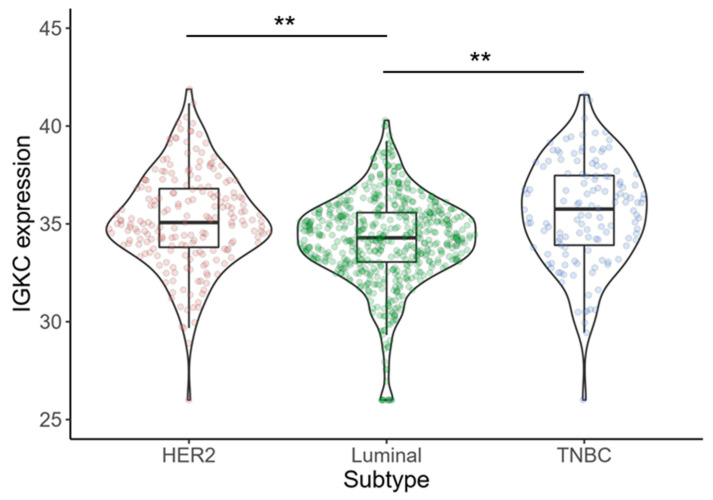
Statistically significant association of *IGKC* expression and molecular subtype (*p* < 0.001; one-way ANOVA). Triple-negative cancers and HER2-positive cancers show higher expression of *IGKC* mRNA compared to luminal cancers. ** *p* < 0.001.

**Figure 3 cancers-13-03626-f003:**
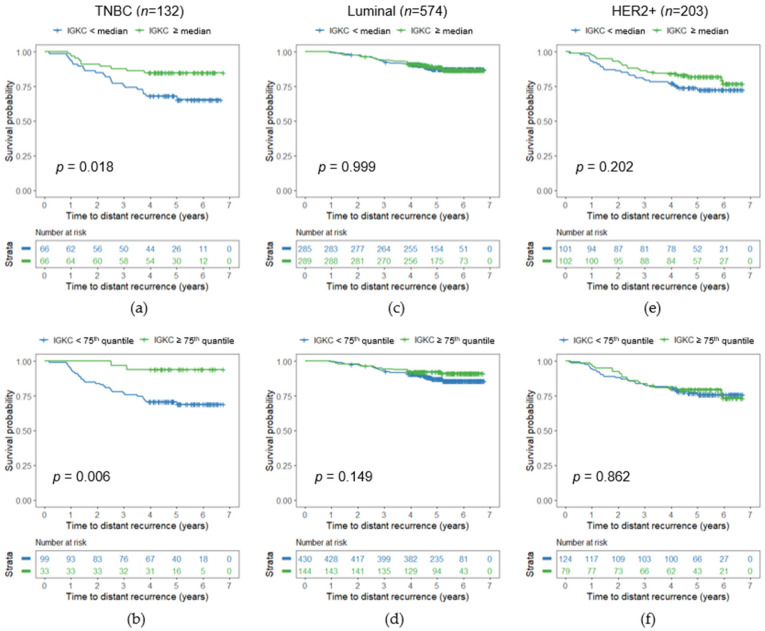
Influence of cancer *IGKC* expression on DDFS illustrated by Kaplan–Meier plots for (**a**,**b**) triple-negative (TNBC), (**c**,**d**) luminal, and (**e**,**f**) HER2+ cancers in the FinHer trial, with *IGKC* dichotomized as low (<median of the corresponding subtype) and high (≥ median of the corresponding subtype) (upper panels) as well as dichotomized as low (<75th quantile of the corresponding subtype) and high (≥75th quantile of the corresponding subtype) (lower panels). The *p*-value from the log-rank test is given in the figure.

**Table 1 cancers-13-03626-t001:** Association of *IGKC* mRNA content in the tumor with clinicopathological characteristics. *IGKC* low was defined as <median (*n* = 452) and *IGKC* high as ≥median (*n* = 457). The *p* value from Fisher’s exact test is given. IGKC, immunoglobulin kappa C; ER, estrogen receptor; PR, progesterone receptor; HER2, human epidermal growth factor receptor 2; TNBC, triple negative breast cancer.

Characteristic		N	*IGKC* Low *n* (%)	*IGKC* High *n* (%)	*p*
Age	<50 years	399	195 (49)	204 (51)	
	≥50 years	510	257 (50)	253 (50)	0.688
Grade ^1^	Grade I	133	73 (55)	60 (45)	
	Grade II	365	202 (55)	163 (45)	
	Grade III	371	158 (43)	213 (57)	0.001
ER status	Positive	250	93 (37)	157 (63)	
	Negative	659	359 (54)	300 (46)	<0.001
PR status ^1^	Positive	382	160 (42)	222 (58)	
	Negative	526	292 (56)	234 (44)	<0.001
HER2 status	Positive	706	367 (52)	339 (48)	
	Negative	203	85 (41)	118 (59)	0.013
Ki67 ^1^	≤20%	412	236 (57)	176 (43)	
	>20%	397	168 (42)	229 (58)	<0.001
Molecular subtype	Luminal	574	321 (56)	253 (44)	
	TNBC	132	46 (35)	86 (65)	
	HER2+	203	85 (41)	118 (59)	<0.001
pT stage ^1^	pT1	381	189 (50)	192 (50)	
	pT2	447	226 (51)	221 (49)	
	pT3 or pT4	80	36 (45)	44 (55)	0.356
pN stage	pN0	96	42 (44)	54 (56)	
	pN1	786	392 (50)	394 (50)	
	pN2 or pN3	27	18 (67)	9 (33)	0.113
Breast surgery	Mastectomy	540	260 (48)	280 (52)	
	Breast conserving	369	192 (52)	177 (48)	0.252
Lymph node dissection ^1^	Axillary dissection	877	435 (50)	442 (50)	
	Sentinel node	31	16 (52)	15 (48)	0.857

^1^ Missing values: grade N = 40, PR status N = 1, Ki67 N = 100, pT stage N = 1.

**Table 2 cancers-13-03626-t002:** Multivariable Cox analysis for DDFS. 868 patients had complete data for all variables and were included in the multivariable analysis. *IGKC* expression was included as a continuous variable. The results of the univariable analysis are shown in Appendix A. CI, confidence interval; IGKC, immunoglobulin kappa C; HER2, human epidermal growth factor receptor 2; TNBC, triple-negative breast cancer.

Variable	Hazard Ratio	95% CI	*p*
*IGKC*	0.930	0.870–0.995	0.034
Age			
≤50 years	1.000		
>50 years	0.899	0.643–1.256	0.532
Molecular subtype			
Luminal	1.000		
TNBC	2.661	1.667–4.249	<0.001
HER2+	2.122	1.425–3.159	<0.001
pT stage			
pT1	1.000		
pT2-4	1.605	1.111–2.319	0.012
pN stage			
pN0	1.000		
pN1-3	4.369	1.993–9.581	<0.001
Grade			
Grade I	1.000		
Grade II–III	2.871	1.235–6.675	0.014

**Table 3 cancers-13-03626-t003:** Univariable and multivariable Cox analyses for DDFS in patients with TNBC. 132 patients had data for *IGKC* and were included in the univariable analysis. 129 patients had complete data for all variables and were included in the multivariable analysis. *IGKC* expression was included as a continuous variable. The corresponding results for *IGKC* dichotomized using the median or top-quartile expression as a cutoff are shown in Appendix A. CI, confidence interval; IGKC, immunoglobulin kappa C; HER2, human epidermal growth factor receptor 2.

Variable	Univariable Analysis			Multivariable Analysis		
	Hazard Ratio	95% CI	*p*	Hazard Ratio	95% CI	*p*
*IGKC*	0.907	0.806–1.022	0.109	0.843	0.724–0.983	0.029
Age						
≤50 years	1.000			1.000		
>50 years	0.427	0.209–0.874	0.020	0.319	0.143–0.708	0.005
pT stage						
pT1	1.000			1.000		
pT2-4	0.839	0.388–1.813	0.654	1.421	0.610–3.310	0.416
pN stage						
pN0	1.000			1.000		
pN1-3	2.796	1.076–7.262	0.035	4.025	1.327–12.209	0.014
Grade ^1^						
Grade I-II ^2^	1.000			1.000		
Grade III	0.642	0.286–1.442	0.283	0.778	0.332–1.823	0.563

^1^*n* = 129; ^2^ grade was dichotomized as I-II vs. III since there was only one TNBC patient with a grade I tumor.

## Data Availability

The dataset analyzed during the current study is available from the corresponding author on reasonable request.

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
