# Peer review of "Prognostic Impact of Immunoglobulin Kappa C (*IGKC*) in Early Breast Cancer"

_cancers, 2021, doi:10.3390/cancers13143626_

Round 1

Reviewer 1 Report

This is an interesting study reporting an association between IGKC and prognosis on a large population from FinHer trial. The paper is well written, the objective and the design are clear. Results are in line with previous literature and provide further evidence on the role of the immune system in triple negative breast cancer. I think the paper would deserve publication.

Author Response

Reviewer #1

Comment 1. This is an interesting study reporting an association between IGKC and prognosis on a large population from FinHer trial. The paper is well written, the objective and the design are clear. Results are in line with previous literature and provide further evidence on the role of the immune system in triple negative breast cancer. I think the paper would deserve publication.

Response 1. We appreciate very much the positive feedback.

Reviewer 2 Report

The paper titled “Prognostic impact of immunoglobulin kappa C (IGKC) in early breast cancer” examine the relevance of immunoglobulin kappa C (IGKC) as an important part of the humoral immune system, in early breast cancer. Author conclude that there is an independent prognostic significance of cancer IGKC content in breast cancer

However, some questions must be clarifying by the authors

  • In the summary there are two different affirmations: line 28“We show that an increased amount of IGKC in the tumor is linked to longer distant metastasis-free survival, especially in patients whose breast cancer does not express hormone re-receptors or human epidermal growth factor receptor-2” that means that patients with high IGKC have a better prognostic with a longer distant metastasis-free survival. But other affirmation say “Since an improved outcome is associated with the presence of tumor-infiltrating IGKC expressing immune cells, this may be a further argument for the use of immunotherapies in these patients” . Can author explain this?

  • If authors make different statistical analysis the results seem in some cases different, them it is possible to use the IGKC as a breast cancer survival, progression or status biomarker??
  • Authors do not see differences in the values of IGKC between different patient’s treatments. But some of this treatments are against immune system. How authors explain this point?

Author Response

Reviewer #2

The paper titled “Prognostic impact of immunoglobulin kappa C (IGKC) in early breast cancer” examine the relevance of immunoglobulin kappa C (IGKC) as an important part of the humoral immune system, in early breast cancer. Author conclude that there is an independent prognostic significance of cancer IGKC content in breast cancer

However, some questions must be clarifying by the authors

Comment 1. In the summary there are two different affirmations: line 28“We show that an increased amount of IGKC in the tumor is linked to longer distant metastasis-free survival, especially in patients whose breast cancer does not express hormone re-receptors or human epidermal growth factor receptor-2” that means that patients with high IGKC have a better prognostic with a longer distant metastasis-free survival. But other affirmation say “Since an improved outcome is associated with the presence of tumor-infiltrating IGKC expressing immune cells, this may be a further argument for the use of immunotherapies in these patients” . Can author explain this?

Response 1. We thank the reviewer for this important question. Interestingly, IGKC is prognostic only in triple-negative breast cancer (TNBC) but not in hormone receptor (HR)-positive breast cancer. Consistent with a stronger impact of the immune system in TNBC are the results of studies with immune checkpoint inhibitors, which are effective in TNBC but not in HR-positive breast cancer [1–3]. One possible explanation for these findings is that TNBC has a higher mutational load and more neoepitopes  [4].

Comment 2.  If authors make different statistical analysis the results seem in some cases different, them it is possible to use the IGKC as a breast cancer survival, progression or status biomarker??

Response 2. This is also a very important question. We focused our analysis on distant disease-free survival (DDFS), which is the purest endpoint for a prognostic biomarker. We show that IGKC is prognostic for DDFS. However, we found no predictive relevance for response to trastuzumab or type of chemotherapy used.

Comment 3. Authors do not see differences in the values of IGKC between different patient’s treatments. But some of this treatments are against immune system. How authors explain this point?

Response 3. We appreciate this comment. At first glance, cytotoxic chemotherapy dampens the immune system. However, cell death induced by chemotherapy leads to the release of neoantigens, which in turn amplify the immune response [5]. In light of this, it is not surprising that we found no predictive effect for the type of chemotherapy used. Similar to pathological factors (e.g., tumor size or nodal status), the immune system, as measured by the IGKC, also contributes to the baseline risk of a TNBC patient. 

References

  1. Tolaney, S.M.; Barroso-Sousa, R.; Keenan, T.; Li, T.; Trippa, L.; Vaz-Luis, I.; Wulf, G.; Spring, L.; Sinclair, N.F.; Andrews, C.; et al. Effect of Eribulin With or Without Pembrolizumab on Progression-Free Survival for Patients With Hormone Receptor-Positive, ERBB2-Negative Metastatic Breast Cancer: A Randomized Clinical Trial. JAMA Oncol. 2020, 6, 1598–1605, doi:10.1001/jamaoncol.2020.3524.
  2. Schmid, P.; Adams, S.; Rugo, H.S.; Schneeweiss, A.; Barrios, C.H.; Iwata, H.; Dieras, V.; Hegg, R.; Im, S.-A.; Shaw Wright, G.; et al. Atezolizumab and Nab-Paclitaxel in Advanced Triple-Negative Breast Cancer. N. Engl. J. Med. 2018, 379, 2108–2121, doi:10.1056/NEJMoa1809615.
  3. Cortes, J.; Cescon, D.W.; Rugo, H.S.; Nowecki, Z.; Im, S.-A.; Yusof, M.M.; Gallardo, C.; Lipatov, O.; Barrios, C.H.; Holgado, E.; et al. Pembrolizumab plus chemotherapy versus placebo plus chemotherapy for previously untreated locally recurrent inoperable or metastatic triple-negative breast cancer (KEYNOTE-355): a randomised, placebo-controlled, double-blind, phase 3 clinical trial. Lancet 2020, 396, 1817–1828, doi:10.1016/S0140-6736(20)32531-9.
  4. Narang, P.; Chen, M.; Sharma, A.A.; Anderson, K.S.; Wilson, M.A. The neoepitope landscape of breast cancer: implications for immunotherapy. BMC Cancer 2019, 19, 200, doi:10.1186/s12885-019-5402-1.
  5. Zitvogel, L.; Apetoh, L.; Ghiringhelli, F.; Kroemer, G. Immunological aspects of cancer chemotherapy. Nat. Rev. Immunol. 2008, 8, 59–73, doi:10.1038/nri2216.
  6. Joensuu, H.; Bono, P.; Kataja, V.; Alanko, T.; Kokko, R.; Asola, R.; Utriainen, T.; Turpeenniemi-Hujanen, T.; Jyrkkio, S.; Moykkynen, K.; et al. Fluorouracil, epirubicin, and cyclophosphamide with either docetaxel or vinorelbine, with or without trastuzumab, as adjuvant treatments of breast cancer: final results of the FinHer Trial. J. Clin. Oncol. 2009, 27, 5685–5692, doi:10.1200/JCO.2008.21.4577.
  7. Laible, M.; Schlombs, K.; Kaiser, K.; Veltrup, E.; Herlein, S.; Lakis, S.; Stöhr, R.; Eidt, S.; Hartmann, A.; Wirtz, R.M.; et al. Technical validation of an RT-qPCR in vitro diagnostic test system for the determination of breast cancer molecular subtypes by quantification of ERBB2, ESR1, PGR and MKI67 mRNA levels from formalin-fixed paraffin-embedded breast tumor specimens. BMC Cancer 2016, 16, 398, doi:10.1186/s12885-016-2476-x.
  8. Varga, Z.; Lebeau, A.; Bu, H.; Hartmann, A.; Penault-Llorca, F.; Guerini-Rocco, E.; Schraml, P.; Symmans, F.; Stoehr, R.; Teng, X.; et al. An international reproducibility study validating quantitative determination of ERBB2, ESR1, PGR, and MKI67 mRNA in breast cancer using MammaTyper®. Breast Cancer Res. 2017, 19, 55, doi:10.1186/s13058-017-0848-z.
  9. Kostadima, L.; Pentheroudakis, G.; Fountzilas, G.; Dimopoulos, M.; Pectasides, D.; Gogas, H.; Stropp, U.; Christodoulou, C.; Samantas, E.; Wirtz, R.; et al. Survivin and glycodelin transcriptional activity in node-positive early breast cancer: mRNA expression of two key regulators of cell survival. Breast Cancer Res. Treat. 2006, 100, 161–167, doi:10.1007/s10549-006-9240-x.
  10. Schmidt, M.; Bohm, D.; Torne, C. von; Steiner, E.; Puhl, A.; Pilch, H.; Lehr, H.-A.; Hengstler, J.G.; Kolbl, H.; Gehrmann, M. The humoral immune system has a key prognostic impact in node-negative breast cancer. Cancer Res 2008, 68, 5405–5413, doi:10.1158/0008-5472.CAN-07-5206.
  11. Schmidt, M.; Hellwig, B.; Hammad, S.; Othman, A.; Lohr, M.; Chen, Z.; Boehm, D.; Gebhard, S.; Petry, I.; Lebrecht, A.; et al. A comprehensive analysis of human gene expression profiles identifies stromal immunoglobulin kappa C as a compatible prognostic marker in human solid tumors. Clin Cancer Res 2012, 18, 2695–2703, doi:10.1158/1078-0432.CCR-11-2210.
  12. Simon, R.M.; Paik, S.; Hayes, D.F. Use of archived specimens in evaluation of prognostic and predictive biomarkers. J. Natl. Cancer Inst. 2009, 101, 1446–1452, doi:10.1093/jnci/djp335.

Reviewer 3 Report

In this manuscript, Schmidt et al. reported an association between mRNA abundance 
of IGKC and distant disease free survival (DDFS) in patients with node positive  or node negative early breast cancer patients in FinHer trial. Moreover, the  association was found mainly in TNBC but not luminal nor Her2-positive sub-
populations. 

This is a clearly-written manuscript. However I have a few concerns. Without further  clarification, I can not recommend publication or revision of this manuscript.

1. This is an add-on analysis of the FinHer trial. As docetaxel and trastuzumab significantly improved DDFS in the original report, systemic treatment should be an important variable to be analyzed in this research. However it was not included here.
2. I am not sure if the DDFS in the HER2+ population is comparable to that of the 
original report.
3. The quality control details for quantitative mRNA analysis is lacking. It is uncertain if the analysis is consistent enough to discern the subtle difference between groups as shown in this report.

Minor comments
1. Has the author done an analysis on PFS of local recurrence?
2. Surgery and radiotherapy details can be added to the demographic table.
3. In the discussion, the authors overinterpreted IGKC mRNA abundance to indicate a role of tumor infiltrating immune cells in TNBC.
4. The authors did not provide sufficient preclinical/clinical/data exploration 
rationale of focusing on IGKC.
5. In Line 142 the author may want to reconsider if the the definition of delta 
cycle threshold is appropriate.

Author Response

Reviewer #3

In this manuscript, Schmidt et al. reported an association between mRNA abundance

of IGKC and distant disease free survival (DDFS) in patients with node positive  or node negative early breast cancer patients in FinHer trial. Moreover, the  association was found mainly in TNBC but not luminal nor Her2-positive sub-populations.

This is a clearly-written manuscript. However I have a few concerns. Without further  clarification, I can not recommend publication or revision of this manuscript.

Comment 1. 1. This is an add-on analysis of the FinHer trial. As docetaxel and trastuzumab significantly improved DDFS in the original report, systemic treatment should be an important variable to be analyzed in this research. However it was not included here.

Response 1. We appreciate the reviewer's comment. However, we have investigated the role of IGKC and presented the findings at the end of the results section of our manuscript:

3.5. Cancer IGKC mRNA content shows no association with systemic treatment effects

An examination of the potential interactions between IGKC expression and the systemic treatments given with DDFS revealed no significant interaction with the type of chemotherapy or whether or not trastuzumab was administered (P interaction = 0.855 and 0.684, respectively).“

Comment 2. 2. I am not sure if the DDFS in the HER2+ population is comparable to that of the

original report.

Response 2. We appreciate this comment. Because we performed our calculations in the database used in the second report of the FinHer trial, the overall survival of HER2-positive patients is unchanged from that report [6]. For illustration, we performed a Kaplan-Meier analysis showing the effect of trastuzumab in HER2-positive patients:

The hazard ratio (HR) of 0.65 corresponds exactly to the original report of the 2009 FinHer study [6]:

Comment 3. 3. The quality control details for quantitative mRNA analysis is lacking. It is uncertain if the analysis is consistent enough to discern the subtle difference between groups as shown in this report.

Response 3. We apologize for this omission. The nucleic acid extraction was done by commercially available kits, whose robustness and reproducibility to detect subtile differences in RNA expression by RT-qPCR has been described previously and led to the approval of e.g. subtyping markers RNXtract® and MammaTyper® as in-vitro diagnostic tool (IVD) for breast cancer [7,8]. The technical development phase included determination of the median fragment leghths of 200 bp by bioanalyzer independent of age of tissue or other preanalytical variables (e.g. time to fixation) before formalin based fixation, which is sufficient for the amplicon lenght of IGKC  (96 bp).

We have added the following sentence: 

The same quality measures were applied as were previously used  for the successful clinical development of RNXtract® and MammaTyper® IVD [7,8]. Each extraction was checked for mRNA quantity and amplificability by using clinically developed reference genes demonstrating sufficient fragment length of total RNA extract for the amplification. Samples with low mRNA content (raw CT value of CALM2 >30) were excluded from analysis. DNA content after DNase digestion was checked by PCR using DNA specific, intron-spanning PAEP gene as quality control gene [9]. Assay performance and specificity of the IGKC assay was validated by using commercially available universal human reference RNA (Stratagene, La Jolla, USA ) and human genomic DNA (Roche Diagnostics, Mannheim, Germany).

Minor comments

Comment1. 1. Has the author done an analysis on PFS of local recurrence?

Response 1. We appreciate the reviewer's suggestion but did not perform an analysis of local recurrence. The aim of our manuscript was to investigate the prognostic significance of IGKC for distant disease-free survival. Regarding local recurrence, surgery and radiotherapy must be taken into account, which might obscure the pure prognostic significance of a biomarker.

Comment 2. 2. Surgery and radiotherapy details can be added to the demographic table.

Response 2. This is an important suggestion. We modified 3.2. and added surgery and radiotherapy:

3.2. IGKC expression is associated with adverse histopathological characteristics

First, we examined an association between cancer IGKC expression (< median vs ≥ median) and clinicopathological characteristics using Fisher’s exact test. High levels of IGKC were significantly associated with poor histological grade of differentiation (P < 0.001), negative ER (P < 0.001) and PR (P < 0.001) status, positive HER2 status (P = 0.013), high (>20%) Ki-67 expression (P < 0.001), and HER2-positive subtype as well as TNBC molecular subtype (P < 0.001). Tumor IGKC mRNA content was not associated with age at diagnosis (P = 0.688), tumor size (P = 0.356),axillary nodal status (P = 0 .113), surgery of the breast (P = 0.252) or lymph node dissection (P = 0.857) (Table 1). Radiotherapy was given according to each institution's guidelines; it was mandated after breast-conserving surgery and was given to most patients with node-positive disease.

Table 1. Association of IGKC mRNA content in the tumor with clinicopathological characteristics. IGKC low was defined as < median (n = 452) and IGKC high as ≥ median (n = 457). The P value from Fisher´s Exact test is given. IGKC, immunoglobulin kappa C; ER, estrogen receptor; PR, progesterone receptor; HER2, human epidermal growth factor receptor 2; TNBC, triple negative breast cancer

Characteristic

N

IGKC low n (%)

IGKC high n (%)

P

Age

<50 years

399

195 (49)

204 (51)

≥50 years

510

257 (50)

253 (50)

0.688

Grade 1

Grade I

133

73 (55)

60 (45)

Grade II

365

202 (55)

163 (45)

Grade III

371

158 (43)

213 (57)

0.001

ER status

Positive

250

93 (37)

157 (63)

Negative

659

359 (54)

300 (46)

<0.001

PR status 1

Positive

382

160 (42)

222 (58)

Negative

526

292 (56)

234 (44)

<0.001

HER2 status

Positive

706

367 (52)

339 (48)

Negative

203

85 (41)

118 (59)

0.013

Ki67 1

≤20%

412

236 (57)

176 (43)

>20%

397

168 (42)

229 (58)

<0.001

Molecular subtype

Luminal

574

321 (56)

253 (44)

TNBC

132

46 (35)

86 (65)

HER2+

203

85 (41)

118 (59)

<0.001

pT stage 1

pT1

381

189 (50)

192 (50)

pT2

447

226 (51)

221 (49)

pT3 or pT4

80

36 (45)

44 (55)

0.356

pN0

96

42 (44)

54 (56)

pN stage

pN1

786

392 (50)

394 (50)

pN2 or pN3

27

18 (67)

9 (33)

0.113

Breast surgery

Mastectomy

540

260 (48)

280 (52)

Breast conserving

369

192 (52)

177 (48)

0.252

Lymph node

dissection 1

Axillary dissection

877

435 (50)

442 (50)

Sentinel node

31

16 (52)

15 (48)

0.857

1 Missing values: grade N=40, PR status N=1, Ki67 N=100, pT stage N=1 

Comment 3. 3. In the discussion, the authors overinterpreted IGKC mRNA abundance to indicate a role of tumor infiltrating immune cells in TNBC.

Response 3. This important comment is very much appreciated. We had shown with multi-gene array analyses that a B cell metagene representing the humoral immune system has an independent positive prognostic role in untreated node-negative breast cancer [10]. At the time we presented our results, some colleagues were unsure whether the transcripts of this metagene originated from immune cells or from breast cancer. The next step was to use receiver-operating characteristic analyses to identify robust markers from our 60-gene, B-cell metagene. We identified IGKC, which as a single marker is similarly predictive and prognostic as the entire B-cell metagene. Using immunohistochemical staining and confocal fluorescence microscopy, we demonstrated that tumor-infiltrating plasma cells are the source of IGKC expression [11]. Considering that tumor-infiltrating plasma cells are a source of IGKC, we do not feel that we have overinterpreted our results.

Comment 4. 4. The authors did not provide sufficient preclinical/clinical/data exploration

rationale of focusing on IGKC.

Response 4. We welcome this comment. However, given the history of IGKC described above, we are confident that it is a valuable biomarker with a solid biological basis in breast cancer. Our early results investigating IGKC were performed using so-called "convenience samples" It is generally accepted that samples from a prospective randomized trial lead to a higher level of evidence [12]. For this reason, we validated our results regarding the prognostic role of IGKC in the FinHer study.

Comment 5. 5. In Line 142 the author may want to reconsider if the the definition of delta

cycle threshold is appropriate.

Response 5. This is an important suggestion. We have adopted the definition from „ΔCt = 40 − [Ct gene of interest− Ct (mean of CALM2)“ to „ΔCt = 40 − [Ct GOI (mean of gene of interest)− Ct REF (mean of CALM2)] to be more precise. Apart from this, we believe that this definition of the delta cycle threshold is appropriate.  We particularly used the 40-DCT method, i.e. subtracting the DCT values from the total number of standard PCR cycling to facilitate ease of interpretation, as thereby higher mRNA expression levels are positively correlated with higher 40-DCT values, whereas they would be negatively associated with DCT values without subtraction as we have done in previous reports [7,8]

References

  1. Tolaney, S.M.; Barroso-Sousa, R.; Keenan, T.; Li, T.; Trippa, L.; Vaz-Luis, I.; Wulf, G.; Spring, L.; Sinclair, N.F.; Andrews, C.; et al. Effect of Eribulin With or Without Pembrolizumab on Progression-Free Survival for Patients With Hormone Receptor-Positive, ERBB2-Negative Metastatic Breast Cancer: A Randomized Clinical Trial. JAMA Oncol. 2020, 6, 1598–1605, doi:10.1001/jamaoncol.2020.3524.
  2. Schmid, P.; Adams, S.; Rugo, H.S.; Schneeweiss, A.; Barrios, C.H.; Iwata, H.; Dieras, V.; Hegg, R.; Im, S.-A.; Shaw Wright, G.; et al. Atezolizumab and Nab-Paclitaxel in Advanced Triple-Negative Breast Cancer. N. Engl. J. Med. 2018, 379, 2108–2121, doi:10.1056/NEJMoa1809615.
  3. Cortes, J.; Cescon, D.W.; Rugo, H.S.; Nowecki, Z.; Im, S.-A.; Yusof, M.M.; Gallardo, C.; Lipatov, O.; Barrios, C.H.; Holgado, E.; et al. Pembrolizumab plus chemotherapy versus placebo plus chemotherapy for previously untreated locally recurrent inoperable or metastatic triple-negative breast cancer (KEYNOTE-355): a randomised, placebo-controlled, double-blind, phase 3 clinical trial. Lancet 2020, 396, 1817–1828, doi:10.1016/S0140-6736(20)32531-9.
  4. Narang, P.; Chen, M.; Sharma, A.A.; Anderson, K.S.; Wilson, M.A. The neoepitope landscape of breast cancer: implications for immunotherapy. BMC Cancer 2019, 19, 200, doi:10.1186/s12885-019-5402-1.
  5. Zitvogel, L.; Apetoh, L.; Ghiringhelli, F.; Kroemer, G. Immunological aspects of cancer chemotherapy. Nat. Rev. Immunol. 2008, 8, 59–73, doi:10.1038/nri2216.
  6. Joensuu, H.; Bono, P.; Kataja, V.; Alanko, T.; Kokko, R.; Asola, R.; Utriainen, T.; Turpeenniemi-Hujanen, T.; Jyrkkio, S.; Moykkynen, K.; et al. Fluorouracil, epirubicin, and cyclophosphamide with either docetaxel or vinorelbine, with or without trastuzumab, as adjuvant treatments of breast cancer: final results of the FinHer Trial. J. Clin. Oncol. 2009, 27, 5685–5692, doi:10.1200/JCO.2008.21.4577.
  7. Laible, M.; Schlombs, K.; Kaiser, K.; Veltrup, E.; Herlein, S.; Lakis, S.; Stöhr, R.; Eidt, S.; Hartmann, A.; Wirtz, R.M.; et al. Technical validation of an RT-qPCR in vitro diagnostic test system for the determination of breast cancer molecular subtypes by quantification of ERBB2, ESR1, PGR and MKI67 mRNA levels from formalin-fixed paraffin-embedded breast tumor specimens. BMC Cancer 2016, 16, 398, doi:10.1186/s12885-016-2476-x.
  8. Varga, Z.; Lebeau, A.; Bu, H.; Hartmann, A.; Penault-Llorca, F.; Guerini-Rocco, E.; Schraml, P.; Symmans, F.; Stoehr, R.; Teng, X.; et al. An international reproducibility study validating quantitative determination of ERBB2, ESR1, PGR, and MKI67 mRNA in breast cancer using MammaTyper®. Breast Cancer Res. 2017, 19, 55, doi:10.1186/s13058-017-0848-z.
  9. Kostadima, L.; Pentheroudakis, G.; Fountzilas, G.; Dimopoulos, M.; Pectasides, D.; Gogas, H.; Stropp, U.; Christodoulou, C.; Samantas, E.; Wirtz, R.; et al. Survivin and glycodelin transcriptional activity in node-positive early breast cancer: mRNA expression of two key regulators of cell survival. Breast Cancer Res. Treat. 2006, 100, 161–167, doi:10.1007/s10549-006-9240-x.
  10. Schmidt, M.; Bohm, D.; Torne, C. von; Steiner, E.; Puhl, A.; Pilch, H.; Lehr, H.-A.; Hengstler, J.G.; Kolbl, H.; Gehrmann, M. The humoral immune system has a key prognostic impact in node-negative breast cancer. Cancer Res 2008, 68, 5405–5413, doi:10.1158/0008-5472.CAN-07-5206.
  11. Schmidt, M.; Hellwig, B.; Hammad, S.; Othman, A.; Lohr, M.; Chen, Z.; Boehm, D.; Gebhard, S.; Petry, I.; Lebrecht, A.; et al. A comprehensive analysis of human gene expression profiles identifies stromal immunoglobulin kappa C as a compatible prognostic marker in human solid tumors. Clin Cancer Res 2012, 18, 2695–2703, doi:10.1158/1078-0432.CCR-11-2210.
  12. Simon, R.M.; Paik, S.; Hayes, D.F. Use of archived specimens in evaluation of prognostic and predictive biomarkers. J. Natl. Cancer Inst. 2009, 101, 1446–1452, doi:10.1093/jnci/djp335.

Reviewer 4 Report

There have been many studies evaluating different expression patterns in breast cancer as a role in prognosis. Nonetheless, the authors showed experience in the direction of the current study and the study is well conducted on a sufficient number of samples, thus making the findings of the current study important to be published.

Figure 2: Please show the distribution of data points by adding a sina and a violin plot to this figure (do not remove the box and wiskers).

Author Response

Reviewer #4

Comment 1. There have been many studies evaluating different expression patterns in breast cancer as a role in prognosis. Nonetheless, the authors showed experience in the direction of the current study and the study is well conducted on a sufficient number of samples, thus making the findings of the current study important to be published.

Figure 2: Please show the distribution of data points by adding a sina and a violin plot to this figure (do not remove the box and wiskers).

Response 1: We thank the reviewer for the generally positive feedback and encouraging comments. We agree with the importance to show the distribution of data points and and added a sina and violin plot to Figure 2:

Figure 2. Statistically significant association of IGKC expression and molecular subtype (p<0.001; one-way ANOVA). Triple-negative cancers and HER2-positive cancers show higher expression of IGKC mRNA compared to luminal cancers. ** P < 0.001 

Round 2

Reviewer 3 Report

Thank you for the nicely organized response. My concerns are mostly reasonably addressed and are updated in relevant texts.

The only remaining minor comment is on the definition of delta Ct. In my understanding, delta Ct refers to "Ct differences" between GOI to normalizer (not "40-Ct differences"). With this definition, it would make sense for the following sentence: "Final expression values were generated by substracting delta Ct from the total number of cycles (40-delta Ct)" . In the original definition, final expression values would have been calculated from 40-(40-Ct(GOI)-CT(normalizer)), which would have been much less than 35.